# Effect of Alkanolamines on the Early-Age Strength and Drying Shrinkage of Internal Curing of Mortars

Dandan Wan [1], Rongjin Liu [1,2,3,*], Tianyu Gao [1], Daiyan Jing [4,*] and Fuhua Lu [1]

1   College of Materials Science and Engineering, Guilin University of Technology, Guilin 541004, China
2   Collaborative Innovation Center for Exploration of Nonferrous Metal Deposits and Efficient Utilization of Resources in Guangxi, Guilin University of Technology, Guilin 541004, China
3   Guangxi Engineering and Technology Center for Utilization of Industrial Waste Residue in Building Materials, Guilin University of Technology, Guilin 541004, China
4   Guangxi Maibu New Material Limited Company, Guilin 541004, China
*   Correspondence: liujin@glut.edu.cn (R.L.); jdy163yx@163.com (D.J.);
    Tel.: +86-181-7832-6268 (R.L.); +86-180-7731-5562 (D.J.)

**Abstract:** In this study, the reduction of the early-age strength of mortar caused by the traditional super absorbent polymer (SAP) was solved. Two types of alkanolamines (Alkanolamine-A and alkanolamine-B) and calcium nitrite were compounded with traditional SAP as early-age strength components and then were added into the mortar to evaluate effects on early-age strength and drying shrinkage. Results showed that adding two types of alkanolamines could significantly improve the early-age compressive strength of mortar. The addition of alkanolamines could refine the pore of cement paste and reduce the average pore size, which can be reflected by the increased strength and improved mortar drying shrinkage. Among them, the mortar with the composite of alkanolamine-A and alkanolamine-B added showed excellent performance. Its mechanical properties increased to 129% at day 3 and 139% at day 7 compared with the reference group; compressive strength can reach up to 31.8 MPa. The average pore size of the cement paste decreased by 27.8%, and the drying shrinkage was significantly smaller compared with the reference group.

**Keywords:** mortar; alkanolamines; internal curing; early-age strength; drying shrinkage

## 1. Introduction

Jensen and Hansen [1] added a super absorbent polymer (SAP) into concrete for internal curing (IC) in 2001. Since then, the role of SAP in the internal curing (IC) of high performance concrete has gained wide recognition. SAP can significantly delay the decrease in the internal moisture content of concrete-based materials and can inhibit self-shrinkage [2,3]. Therefore, it can be used in the internal maintenance of concrete. However, SAP can also release water and create pores during the process of hydration. The created pores have a negative effect on the strength development of concrete [4], especially on the early mechanical properties.

Super absorbent polymer (SAP) is a polymer that can be obtained by the polymerization of monomers containing a multitude of hydrophilic groups (e.g., carboxyl (-COOH), hydroxyl (-OH), and amide groups (-CONH$_2$) [5,6]. It has a three-dimensional network structure, which is formed by a low degree of cross-linking. It also has excellent salt and alkali resistance and excellent water absorption-release properties [7]. By releasing the pre-absorbed water during hydration, SAP added to concrete enables the hydration of cement, keeps the internal humidity of concrete and promotes further hydration of cement [8]. Compared with other materials for internal curing, SAP has two unique properties: it can not only keep the internal humidity of concrete and inhibit shrinkage [9], but can also increase the resistance to frost erosion and improve the durability of concrete by the increased porosity and air content inside the concrete with the water released from

it [10,11]; moreover, the SAP water absorption multiplier can be designed in different situations so as to adapt to the shrinkage under different conditions introduced by different water-to-cement ratio concrete materials [12].

Autogenous shrinkage and drying shrinkage are the direct consequences of the decrease in internal relative humidity (IRH) due to autogenous shrinkage and external drying [13]. SAP can be used to inhibit concrete autogenous and drying shrinkage. Shen et al. [8] studied the effect of different amounts of SAP on the shrinkage and cracking resistance of concrete, by the concrete restraint ring test, and the results showed that the shrinkage value of concrete decreased and concrete cracking time increased with the increase in SAP, indicating a positive effect of SAP on shrinkage restraint. Shi et al. [14] studied the effect of the particle sizes of SAP internal curing materials on the performances of self-compacting concrete. It was considered that both the relative humidity inside the cement and the early autogenous shrinkage of self-compacting concrete (SCC) will decrease with the increase in the introduced amount of internal curing water. Incorporation of highly absorbent polymer SAP can effectively regulate the decrease in IRH of cementite due to self-shrinkage, thus reducing capillary stress and the self-shrinkage of concrete. Zhang et al. [15] pointed out that SAP alone delayed the hydration of cement and increased the percentage of macropores above 100 nm, and introducing lime-type expansion agent (KEA) into SAP mortar could promote the formation of micropores and increase the compressive strength slightly. After regression analysis, a relationship was found between the microscopic volume of pores above 50 nm and the macroscopic compressive strength. The compressive strength decreases gradually with the increase in the percentage of pore sizes above 50 nm.

Super absorbent polymer (SAP) with high water absorption can be used in internal curing of concrete with a low water-to-cement ratio [16,17]. It can release water and create pores as the cement hydrates [18]. The pores can adversely affect the strength of concrete. Tan et al. [19] stated that both the particle size and powder SAP have significant influences on the performances of cementitious materials. When the particle sizes were 140 and 200 mesh, the prepared cementitious materials had higher 28 day compressive strength than others. However, the effect of powdered SAP on the early-age strength of mortar was not mentioned. Researchers have noted that the addition of SAP reduces the mortar porosity via microCT [20]. Lai et al. [21] pointed out that the types of SAP, the mixing amount and the mixing method all affect free water and defects in mortar, which in turn, affects mechanical properties of mortar. When the SAP was added in the form of dry powder (with the same SAP-adding amount), its adverse effect on early-age strength was more significant.

An early-strength accelerant can significantly improve the early setting and hardening of concrete and rapidly improve early mechanical properties of concrete [22]. There are three major categories of traditional early-age strength accelerants: organic small molecules, inorganic salts, and composite early-age strength accelerants [23]. The effects of different early-age strength agents on cement-based materials are different, and they can be concluded in four aspects, namely providing crystalline nuclei for cement hydration, increasing pH value of the cement paste, speeding up the dissolving of cement components and forming complexes with hydration products of $Ca(OH)_2$ [24]. Each of them can speed up early cement hydration reactions and improve early mechanical properties of cementitious materials.

Traditional alkanolamines are TEA, TIPA and so on. Some studies [25] have shown that TEA can enhance the early strength of mortar, and TIPA can enhance the later strength of mortar. When alkanolamine is used as a chemical additive, it chemically enhances the mineral hydration in the cement; thus, it is considered to be added to the mortar in combination with SAP to improve early-age strength.

The purpose of this paper is to find a way to inhibit the decrease in early-age strength of internal curing of mortar without affecting the crack resistance of SAP itself. In order to achieve this requirement, SAP as modified using different types of alkanolamines to

form composite early-age strengthen internal curing agents. Their effects on the early-age strength and drying shrinkage of the mortar were also observed. The early-age strength can be improved by adding early-strength components to promote cement hydration.

## 2. Materials and Methods

### 2.1. Materials

P·O42.5 grade ordinary silicate cement was supplied from Xing'an Conch Cement Co. (Guilin, China). Its density is 3150 kg/m$^3$. The main chemical composition and performance parameters are shown in Tables 1 and 2. Alkanolamine-A and alkanolamine-B were produced by Xilong Science Co., Ltd., with purity over 98%. Calcium nitrite was produced by Shanghai Yi'en Chemical Technology Co. Super absorbent polymer (SAP) was made by our team [26]. The main component is a three-dimensional network structure consisting of acrylamide and 2-acrylamido-2-methylpropanesulfonic acid grafted with tapioca starch. The super absorbent polymer can absorb 87 g/g of the synthetic pore solution and turn into a kind of gel after dilution with tap water. In this experiment, the super absorbent polymer can absorb liquid with 4 times its own weight, i.e., 1 kg of the SAP absorbed 4 kg of tap water. The sand used in this experiment was from Xiamen ISO Standard Sand Co., Ltd. (Xiamen, China), which is a kind of ISO standard sand. The water used was tap water. During the preparation process, a high-speed rotary cutting machine was used, which can make the SAP disperse better and absorb water more quickly to reach the saturation state.

**Table 1.** Chemical composition of cement.

| CaO | SiO$_2$ | Al$_2$O$_3$ | Fe$_2$O$_3$ | MgO | SO$_3$ | Loss |
|---|---|---|---|---|---|---|
| 64.13 | 19.65 | 5.16 | 3.69 | 1.60 | 2.64 | 0.3 |

**Table 2.** Physical properties of cement.

| Supercific Surface Area (m$^2$/kg) | Water Requirement of Normal Consistency (%) | Setting Time (min) | | Flexural Strength (MPa) | | Compressive Strength (MPa) | |
|---|---|---|---|---|---|---|---|
| | | Initial | Final | 3 day | 28 day | 3 day | 28 day |
| 378 | 25.00 | 168 | 205 | 5.9 | 8.3 | 28.2 | 47.0 |

### 2.2. Methods

#### 2.2.1. Mortar Strength

Mortar forming and testing methods refer to Chinese National Standard GB/T 17671-2021. The mortar mixture was loaded into the 40 × 40 × 160 mm molds and cured under ambient temperature of 20 ± 2 °C and a relative humidity of 95%. Mortar mix proportion is in Section 3.

#### 2.2.2. Mortar Drying Shrinkage

Drying shrinkage experiment was determined according to building materials industry standards of China (JC-T603-2004). Drying shrinkage experiment mixture dosing percentage is presented in Table 3. The weight ratio of cement to sand was 1:2, and the water-to-cement ratio was 0.5. The test mold was a prism of 25 × 25 × 280 mm, with copper nails pre-buried on both sides. The specimens were put into a standard maintenance room after molding and then put into the dry maintenance room with a temperature of (20 ± 2) °C and humidity of (60 ± 5)% for maintenance. The drying shrinkage values of the specimens after 3, 7, 14, 21, and 28 days were tested. Cement comparator was used to

measure drying shrinkage, which consists of a micrometer, stand and calibration bar. The specific calculations are as follows:

$$S_{28} = \frac{(L_0 - L_{28}) \times 100}{250} \tag{1}$$

where:

　　$S_{28}$—Drying shrinkage values at day 28, unit is %;
　　$L_0$—Initial measured values, unit is mm;
　　$L_{28}$—Measured values at day 28, unit is mm;
　　250—Effective length of samples, unit is mm;

**Table 3.** Mortar mix proportion design.

| Sample | Number | Cement/g | Standard Sand/g | SAP/g | Water/g | Alkanolamine-A/% * | Calcium Nitrite/% * | Alkanolamine-B/% * |
|---|---|---|---|---|---|---|---|---|
| I | A | 450 | 1350 | 18 | 210 | | | |
| II | B1 | 450 | 1350 | 18 | 210 | 0.05 | | |
| | B2 | 450 | 1350 | 18 | 210 | | 1 | |
| | B3 | 450 | 1350 | 18 | 210 | | | 0.04 |
| III | C1 | 450 | 1350 | 18 | 210 | 0.05 | 1 | |
| | C2 | 450 | 1350 | 18 | 210 | 0.05 | | 0.04 |
| | C3 | 450 | 1350 | 18 | 210 | | 1 | 0.04 |
| | C4 | 450 | 1350 | 18 | 210 | 0.05 | 1 | 0.04 |

* The mass of alkanolamine-A, alkanolamine-B and calcium nitrite are by mass of cement.

### 2.2.3. SEM and XRD

The cement paste samples were prepared following the mix ratio in Table 3 and then cured for 7 days. After that, a small amount of the samples with 3~5 mm diameter size were analyzed by SEM. S4800 Scanning Electron Microscope produced by Nippon Electron was used for SEM analysis. The other samples were grinded and sieved to 74 μm, and the powder samples were used for XRD. X'Per PRO X-ray diffraction analyzer (PANantical, The Netherlands) was used for X-ray diffraction for the samples with a scanning speed of 12°/min and a scanning range of 10° to 80°. The experimental results were analyzed by MDI Jade software.

### 2.2.4. Pore Structure Test

AutoPore V 9600 was used for the pore analysis of the samples. As with SEM, after curing to the named age, samples were taken from the center of the specimen block for mercury pressure testing. Two to three grams was needed for one sample. The total porosity, pore size distribution and other related parameters of the hardened cement paste were tested.

### 3. Mortar Mix Proportion Design

The ratio of cement to sand in this test mortar was 1:3 (mass ratio), and the water-to-cement ratio was 0.5. The water absorption multiplier of the high-absorbent polymer internal maintenance material determined after preliminary laboratory work was four times. The optimal amount of a single early-age strength component based on the effect of the preliminary single early-age strength component on the early-age strength of the polymer mortar was determined. Alkanolamine-A, alkanolamine-B and calcium nitrite doses were 0.05%, 0.04% and 1%, respectively (by mass of cement). Group I was the reference group. Group II was used to study the effect of a single early-age strength component on the early-age strength and drying shrinkage of highly absorbent polymer mortar. Group III was used to study the effect of multi early-age strength components on the early-age strength and drying shrinkage of mortar.

## 4. Results

### 4.1. Effect of SAP on the Early Mechanical Properties of Mortar

From Figures 1 and 2, it can be seen that adding either a single early-age strength component or composite early-age strength components contributed to the strength. Although they belonged to different types of early-age strength components, they all shortened the cement hydration induction period, promoted the cement hydration reaction process, and improved the early-age strength of mortar [24]. After compounding the high-absorbent polymer with single early-age strength components, both the 3- and 7-day compressive strength ratios of the final product increased first and then decreased. Among them, the II-B2 group (mixed with single early-age strength components calcium nitrite) showed excellent compressive performance. The 3-day compressive strength of the II-B2 group could reach 21.1 MPa, which increased to 16.5% compared with the 3-day compressive strength of the reference group. Meanwhile, the 7-day compressive strength of the II-B2 group could reach 31.8 MPa, which had a 24.2% increase compared with the reference group. Whether alkanolamine or calcium nitrite was added to the mortar, the compressive strength ratio of the mortar at 3 and 7 days showed an increasing trend. As for the mortars with multiple components added, the 3-day compressive strength of the III-C1 group (with alkanolamine-A and calcium nitrite added) reached 24.0 MPa, which had a 32.6% increase compared with the reference group. The 7-day compressive strength of the III-C4 group (with alkanolamine-A, calcium nitrite and alkanolamine-B added) reached 37.8 MPa, which had a 47.6% increase compared with the reference group (25.6 MPa).

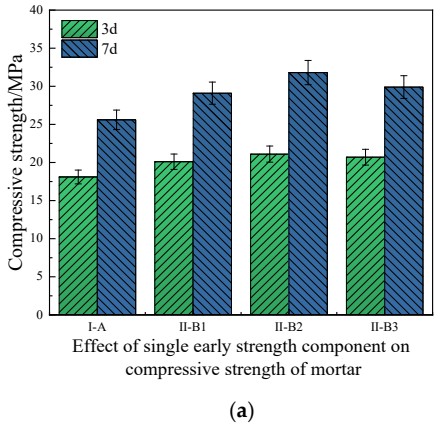

(**a**)

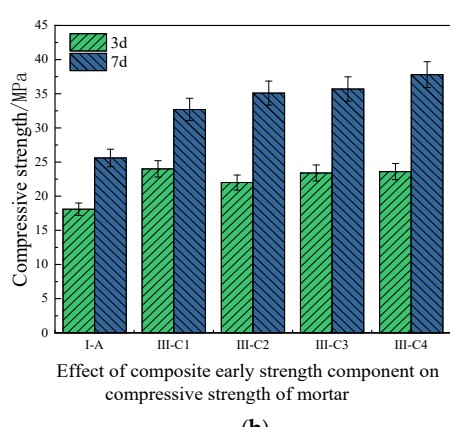

(**b**)

**Figure 1.** Effect of single premature strength component and composite early-age strength component on mortar early-age compressive strength: (**a**) single early-age strength component; (**b**) compound early-age strength components.

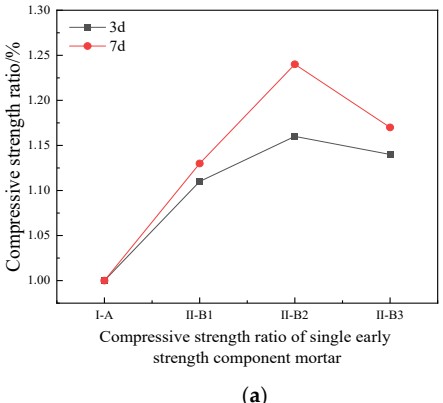

(**a**)

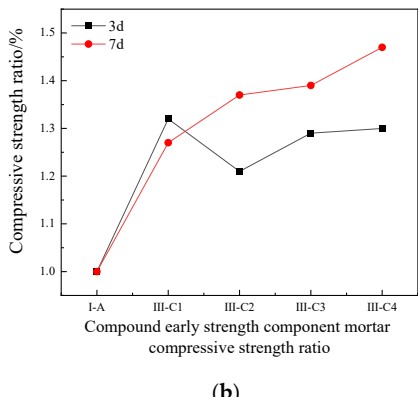

(**b**)

**Figure 2.** Effect of SAP on the early-age compressive strength ratio of mortar: (**a**) single early-age strength component; (**b**) compound early-age strength components.

### 4.2. Effect of SAP on the Drying Shrinkage of Mortar

From Figure 3, it can be seen that the drying shrinkage value of the polymer mortar grew fast at the early stage and slowed down gradually. The overall trend increased gradually. Adding early-strength components can play a positive role in the drying shrinkage of the absorbent polymer mortar. In the experiments with single early-age strength components added, the drying shrinkage values of the polymer mortar with calcium nitrate or alkanolamine-B added were significantly smaller than that of the reference group throughout the maintenance age. Meanwhile, the drying shrinkage of the polymer mortar with the alkanolamine-B added was the smallest. The early drying shrinkage values of the mortar added with alkanolamine-A were slightly higher than the reference group at the beginning, and it was lower than that of the reference group from 14 days onward. The drying shrinkage values of all groups in the experiments with alkanolamine-A early-age strength components added were smaller than that of the reference group. As for the mortars with multiple components added, the group with alkanolamine-A and alkanolamine-B added had the smallest shrinkage of polymer mortar and the smallest drying shrinkage value of mortar. This result was caused by two factors. One was the added early-strength components, which could make the cement paste finish expand in volume at the early stage of cement hydration. The other was the SAP, which could release the water stored inside at the late stage of cement hydration, thus keeping the internal relative humidity and ensuring continuous cement hydration. These two factors interacted with each other, playing positive roles in the drying and shrinkage performance of the mortar [8].

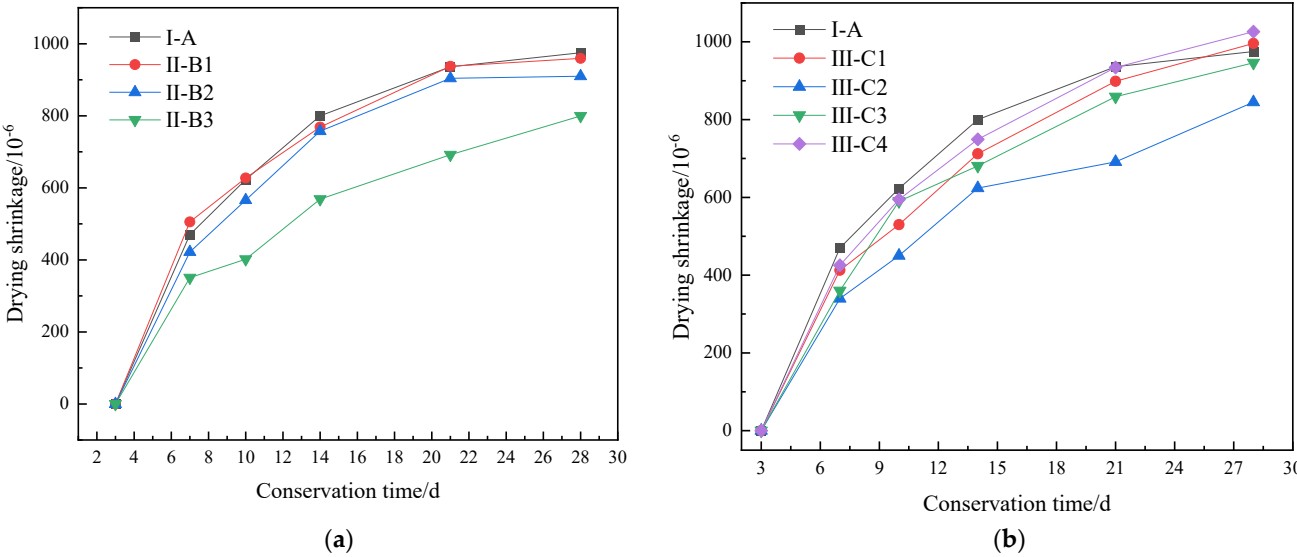

**Figure 3.** Effect of SAP on the drying shrinkage of the polymer mortar: (**a**) effect of single early-age strength component on drying shrinkage of mortar; (**b**) effect of compound early-age strength components on drying shrinkage of mortar.

### 4.3. Micro-Mechanical Analysis

4.3.1. Analysis of the SAP on the Pore Structure of Cement Net Slurry

From Figure 4 and Table 4, it can be seen that the porosity and average pore size of the II-B1 with alkanolamine-A alone added were increased, compared with those of the reference group. This means that the added alkanolamine-A generated a large amount of AFm, which could absorb $SO_4^{2-}$ in the system and convert it into AFt in the late stage of hydration, resulting in volume expansion and causing changes in the pore structure of the slurry system. The other two early-age strength components also had good refining effects on the pore structure. The average pore size of the final product with calcium nitrite and alkanolamine-B added decreased by 9.47% and 18.55%, respectively, compared with the reference group. The average pore size of the cement paste with the composite

containing alkanolamine-A and alkanolamine-B added decreased by a minimum of 27.8%, compared with the reference group. As shown in Table 4, with the addition of the composite containing alkanolamine-A and alkanolamine-B, group III-C2 had the largest percentage of pore sizes between 10 and 50 nm (70.06%). Mehta, P.K. [27] pointed out that gel pores with sizes less than 10 nm are harmless to cementitious materials. Small capillary pores from 10 to 50 nm had a larger effect on shrinkage and creep, while the capillary pores larger than 50 nm had a much greater effect on strength and permeability. Therefore, the III-C2 group with the composite added had a greater effect on mortar shrinkage. This is consistent with the conclusion reached in Section 4.2. The composite containing the early-strength component and high absorbent polymeric internal curing material could play an important role in refining the pore size of cement paste and improving the drying shrinkage of the mortar.

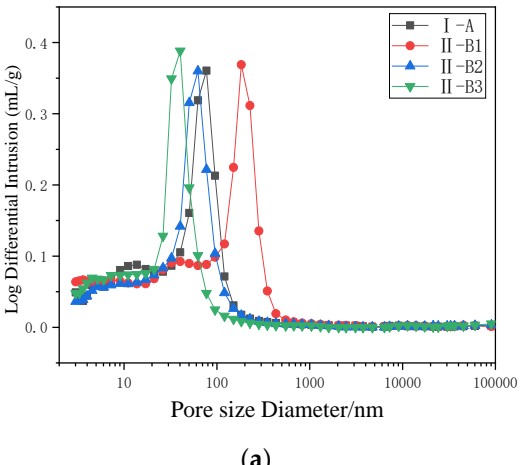

(**a**)

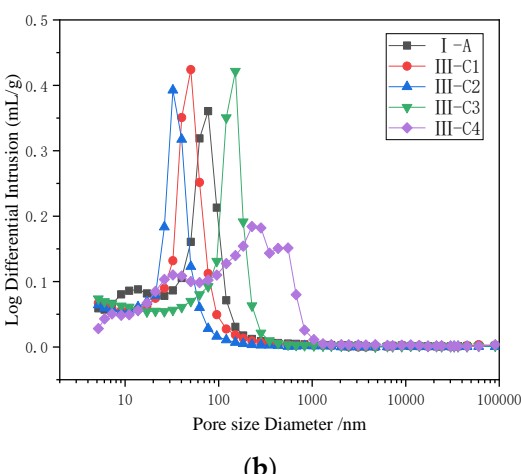

(**b**)

**Figure 4.** Effect of composite premature strength internal maintenance material on 7-day pore size distribution of pure cement paste: (**a**) single early-age strength component; (**b**) compound early-age strength components.

**Table 4.** Effect of composite premature strength inner sample guard material.

| Group | Porosity/% | Average Pore Size/nm | 10~50 nm Aperture Ratio | >50 nm Aperture Ratio |
|---|---|---|---|---|
| I-A | 29.56 | 21.02 | 41.19 | 44.67 |
| II-B1 | 32.54 | 21.71 | 44.26 | 35.07 |
| II-B2 | 31.17 | 19.03 | 51.14 | 32.86 |
| II-B3 | 30.52 | 17.12 | 67.37 | 12.06 |
| III-C1 | 28.79 | 18.65 | 64.01 | 19.85 |
| III-C2 | 27.86 | 15.17 | 70.06 | 10.45 |
| III-C3 | 29.34 | 17.79 | 68.95 | 12.53 |
| III-C4 | 36.41 | 25.44 | 32.53 | 52.8 |

4.3.2. XRD Analysis of Single Early-Age Strength Component Cement Paste

The net cement paste samples were prepared following the mix ratio in Table 3, were then cured for 7 days, and then tested by XRD. The experimental results are shown in Figure 5.

As shown in Figure 5, the strongest diffraction peaks of $Ca(OH)_2$ (d = 0.49170 nm, $2\theta$ = 18.026°) in II-B1 and II-B2 with early-strength components added are significantly lower than that of the I-A benchmark group. Therefore, it can be speculated that early-age strength components II-B1 and II-B2 could consume $Ca(OH)_2$ in the cement paste, shorten the hydration induction period and speed up the cement hydration procedure. It can also be seen that the diffraction intensity of the main mineral phases of cement hydration ($C_3S$, $C_2S$) (d = 0.30672 nm, $2\theta$ = 29.089°) is lower than that of the reference group. This is due to

the decrease in $Ca^{2+}$ concentration in the region. The ion concentration difference of the $C_3S$ hydration inclusions increased, and the osmotic pressure increased as well [28], which resulted in the rupture of hydration inclusions, thus accelerating the hydration of the main mineral phases of cement hydration ($C_3S$, $C_2S$). The early-strength component of the II-B2 was calcium nitrite, which is a kind of inorganic salt. It could promote the generation of the cement hydration product $Ca(OH)_2$, thus speeding up the cement hydration process. Thus, the intensity of the $Ca(OH)_2$ diffraction peak in II-B2 did not decrease significantly compared with that of the reference group. The diffraction peak intensities of the main hydration products can be derived from the MDI Jade analysis, as shown in Table 5.

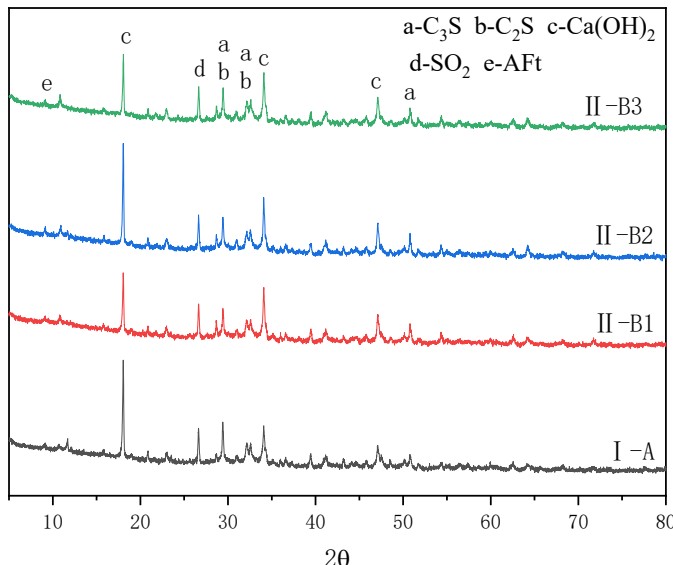

**Figure 5.** Seven-day XRD analysis of cement paste mixed with a single early-age strength component.

**Table 5.** Comparison of hydration degree of 7-day cement paste.

| Diffraction Peak Position | I-A | II-B1 | II-B2 | II-B3 |
|---|---|---|---|---|
| $d_{CH}$ = 0.49170 nm | 2574 | 1181 | 2585 | 1116 |
| $d_{C_3S,C_2S}$ = 0.30672 nm | 728 | 471 | 479 | 478 |
| $d_{Aft}$ = 0.98697 nm | 30 | 39 | 91 | 71 |

From Table 5, it can be seen that the intensity of the $Ca(OH)_2$ diffraction peaks in group II-B1 and group II-B3 decreased by 54.11% and 56.64%, respectively, compared with the reference group. The early-age strength component alkanolamine-A in group II-B1 consisted of alcohol-amine groups, which could form water-soluble complexes with the metal cations in the cement paste [11]. It promoted aluminate phase hydration, thus increasing the early-age strength of the cement paste. The early-age strength of group II-B3 doped with alkanolamine-B is a mechanism that promotes hydrolysis and scattering of the refractory ferrates to achieve early-age strength [29]. The early-age strength component alkanolamine-B in group II-B3 could promote hydrolysis and the scattering of the refractory ferrates to achieve early-age strength. The intensity of the $Ca(OH)_2$ diffraction peak in group II-B2 was higher than that in the reference group. It can be seen from Table 5 that adding early-age strength components could significantly reduce the intensity of the main minerals ($C_2S$, $C_3S$), compared to that of the benchmark group. Thus, it can be concluded that although the mechanisms for different early-age strength components are different, they all speed up the hydration of the cement and promote the generation of cement hydration products. The macroscopic performances of the higher early-age strength of cement mortar were better than those of the c group.

### 4.3.3. Micromorphology Analysis

A microscopic morphological analysis of the 7-day hydration products of single early-strength component polymer mortars was performed by SEM.

From Figure 6a, it can be seen that the generation of the C-S-H gel from the I-A reference group mortar after 7-day hydration was significantly less than the other groups. Individual fibrous C-S-H gels could be seen almost everywhere and they were not interconnected to form honeycomb C-S-H gels. This is evidence of the decrease in the early-age strength of mortar led by an internal curing agent (ICA). The morphology of the II-B1, the high-absorbent polymeric inner-care mortar with alkanolamine-A incorporated for 7 days, is shown in Figure 6b. It can be seen that a petal-like hydrated calcium sulfoaluminate AFm was formed. This was caused by the water-soluble complexes between the unshared electron pairs of the N atoms on the surface of alkanolamine-A and the metal ions in the cement paste, which sped up the dissolving of $C_3A$ and $C_4AF$ and the hydration reaction and improved the early denseness and strength of the cement mortar. From Figure 6c, it can be seen that large amounts of honeycomb C-S-H gel were generated in the II-B2 group, and the generation amount and cross-linking degree were obviously higher than those of the reference group mortar. Adding nitrite could speed up the hydration of the cement and make the mortar more compact. From Figure 6d, it can be seen that the amount of C-S-H gel in the II-B3 group was more than that of the reference group mortar I-A. However, the C-S-H generation was less than that of the polymer mortar group II-B2 with the addition of calcium nitrate. Therefore, it can be concluded that adding alkanolamine-B could have a certain early-age strengthening effect on the high-absorbent polymer mortar, but the effect was not that obvious compared with calcium nitrate.

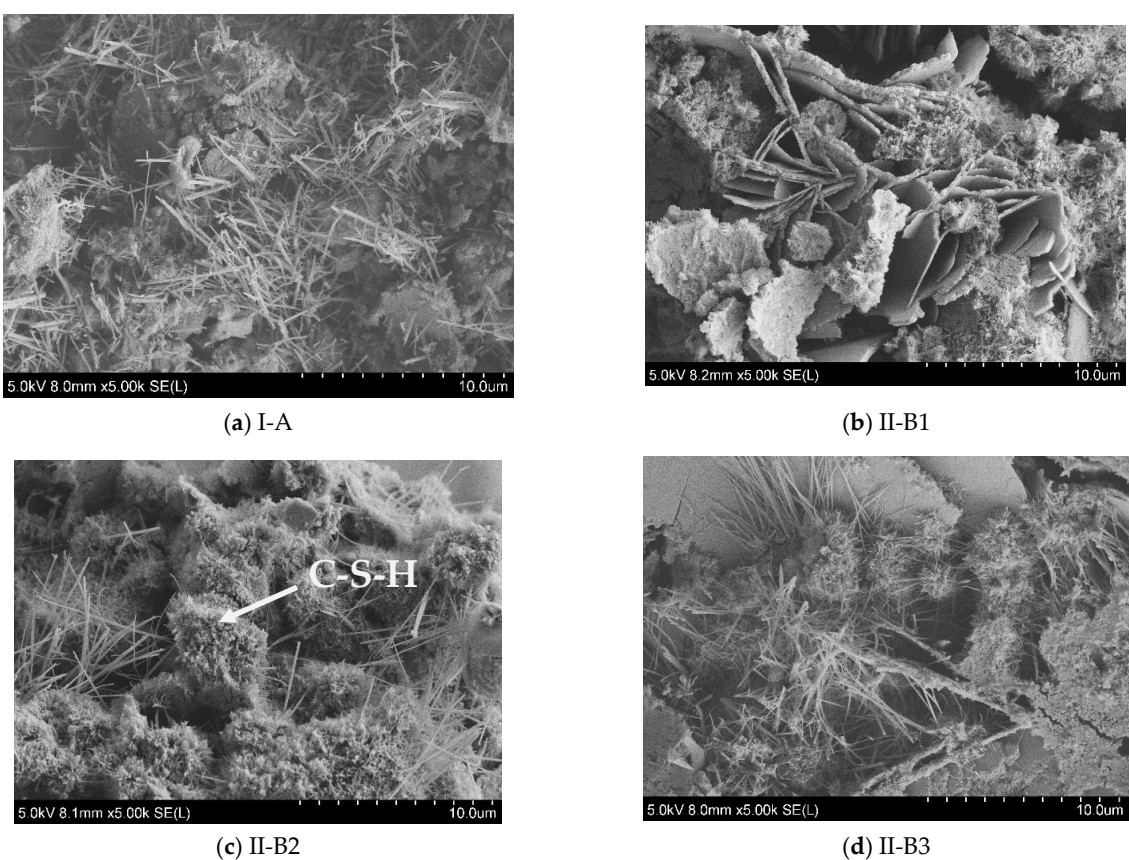

(**a**) I-A

(**b**) II-B1

(**c**) II-B2

(**d**) II-B3

**Figure 6.** SEM electron micrographs of hydration products at 7 days: (**a**) reference group mortar after 7-day hydration; (**b**) mortar with alkanolamine-A incorporated after 7-day hydration; (**c**) mortar with calcium nitrate after 7-day hydration; (**d**) mortar with alkanolamine-B after 7-day hydration.

## 5. Conclusions

This article provides a way to solve the problem of reducing the early-age strength of internal curing of mortar as well as providing technical reference for the applications of SAP in the projects. Based on the results of this study, the following conclusions can be drawn:

1. By adding two types of alkanolamines, we effectively inhibited the early-age compressive strength of mortars without affecting the crack resistance of SAP itself.
2. Cement pastes with different alkanolamines mixed had higher major products (AFt) than the reference group mortar. The diffraction peaks of the main constituent minerals ($C_3S$, $C_2S$) were significantly lower than those of the reference group. It can also be concluded that the alkanolamines promoted cement hydration and sped up water hydration, although their influences and mechanisms are different.
3. Take the effect of composite early-strength internal curing materials on the early mechanical properties of mortar and drying shrinkage into comprehensive consideration, the early performance of the mortar with alkanolamine-A and alkanolamine-B added was the best. Its 3- and 7-day strengths reached 23.4 and 35.7 MPa, respectively. The pore size of 10–50 nm of the cement paste reached 70.06%. The average pore size was 15.17 nm, which had a 27.8% decrease compared with that of the reference group.

**Author Contributions:** Conceptualization, D.W.; data curation, T.G. and D.W.; writing—original draft preparation, D.W. and T.G.; writing—review and editing, D.W., R.L., T.G., D.J. and F.L.; supervision, R.L.; project administration, R.L.; funding acquisition, R.L. All authors have read and agreed to the published version of the manuscript.

**Funding:** The authors are grateful for the funding provided by the National Key Research and Development Program of China (project 2019YFC1906202), Guangxi Key Research and Development Plan (Guike AB19259008, Guike AB20297014).

**Institutional Review Board Statement:** Not applicable.

**Informed Consent Statement:** Not applicable.

**Data Availability Statement:** Not applicable.

**Conflicts of Interest:** The authors declare no conflict of interest.

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
