# Peer review of "Effect of Alkanolamines on the Early-Age Strength and Drying Shrinkage of Internal Curing of Mortars"

_applsci, doi:10.3390/app12199536_

Round 1

Reviewer 1 Report

This paper brings a study that solves the reduction of the initial compressive strength of the mortar caused by the traditional superabsorbent polymer (SAP). However, some adjustments must be made to the paper to be published. Suggestions for improvements can be viewed in the attached file.

Reviewer 2 Report

The authors have investigated the effect of alkanolamines on the early-age strength and drying shrinkage of internal curing of mortars. For this two types of alkanolamines(Alkanolamine-A and alkanolamine-B) and calcium nitrite were compounded with traditional SAP as early-age strength components. However, the authors have not discussed the background of this study why they have chosen these materials. What about the cost? Is it comparable the cost with the increased strength? In addition, the following comments must be considered in revising the manuscript.

Page 5, lines 162-165 not related to the manuscript and must be deleted.

In line 128 and 148 it is given two different cement to mortar ratio. Which one is correct and if both ratios are used, then explain the reason behind it.

Line 169-171: For cement hydration process, the authors have referred reference 23 which dealt with different material than the materials used in the current study. How the authors confirmed same hydration mechanism happened here even though the materials are not same.

It is not clear how the compressive strength ratio in Fig 2 was calculated and how it works/helps? Please explain briefly.

I am not clear about the unit of drying shrinkage values shown in Fig 3.

Throughout the manuscripts English should be improved both for typo and grammatical error. Similarly, font type is also not consistent. See the captions for Fig 5 and 6 which seems different font than others.

Reviewer 3 Report

The present work studies the effect of alkanolamines on mortar. The current manuscript has many typos and redaction issues that should be assessed before publication. My recommendation is for major revision after correcting and considering several points:

1.   The abstract should be rewritten to show the best work results with the alkanolamines.

2.   The meaning of SCC in line 60 is missing.

3.   A reference should be added in lines 90-92.

4.   Over the MS, the letter d appears as a unit of time; it’s better to add the complete word “day”.

5.   In evaluating the mechanical properties and effects of SAP in the mortars, it’s essential to indicate the significant difference and show it in the graphics to make a proper analysis. 

6. Please rewrite the sentences: “no matter what was added inside (alkanolamine-A, alkanolamine-B or calcium nitrite), the 3 d and 7 d compressive strength ratios showed an increasing trend” lines 179-180 and “The early shrinkage value of polymer mortar with alkanolamine-A added was slightly higher than that of the reference group at the beginning” lines 198-199 

7.   The section effect of SAP on the drying shrinkage of mortar is not clear in the analysis of the results. It’s essential to make it easier for the reader to understand the discussion, so this section should be reframed.

8.   It’s necessary to add a reference in lines 205-210.

9.   In line 233, “reached in 4.2” should be replaced by “reached in section 4.2”.

10. What is the relevance of the increase of the diffraction signals intensity in the XRD analysis? lines 250-252, 262-264.

11. The conclusions should be presented in a critical way to reflect the main results of the work.

Round 2

Reviewer 1 Report

Thank you for being invited to review the paper "Effect of Alkanolamines on the Early-age Strength and Drying Shrinkage of Internal Curing of Mortars". The authors adjusted the paper according to the recommendations. Therefore, I consider the paper suitable for publication.

Author Response

We appreciate for reviewers’ warm work earnestly

Reviewer 2 Report

The authors have revised the manuscript in satisfactory level. I have no more comments.

Author Response

(The authors gave the same response as above.)

Reviewer 3 Report

The proper comments have been addressed properly. 

Author Response

(The authors gave the same response as above.)
